# Antioxidant Genetic Profile Modifies Probability of Developing Neurological Sequelae in Long-COVID

**DOI:** 10.3390/antiox11050954

**Published:** 2022-05-12

**Authors:** Marko Ercegovac, Milika Asanin, Ana Savic-Radojevic, Jovan Ranin, Marija Matic, Tatjana Djukic, Vesna Coric, Djurdja Jerotic, Nevena Todorovic, Ivana Milosevic, Goran Stevanovic, Tatjana Simic, Zoran Bukumiric, Marija Pljesa-Ercegovac

**Affiliations:** 1Faculty of Medicine, University of Belgrade, 11000 Belgrade, Serbia; ercegovacmarko@gmail.com (M.E.); masanin@gmail.com (M.A.); ana.savic-radojevic@med.bg.ac.rs (A.S.-R.); njranin@gmail.com (J.R.); marija.matic@med.bg.ac.rs (M.M.); tatjana.djukic@med.bg.ac.rs (T.D.); vesna.coric@med.bg.ac.rs (V.C.); djurdja.jovanovic@med.bg.ac.rs (D.J.); ivana.milosevic00@gmail.com (I.M.); goran_drste@yahoo.com (G.S.); tatjana.simic@med.bg.ac.rs (T.S.); 2Clinic of Neurology, Clinical Centre of Serbia, 11000 Belgrade, Serbia; 3Clinic of Cardiology, Clinical Centre of Serbia, 11000 Belgrade, Serbia; 4Institute of Medical and Clinical Biochemistry, 11000 Belgrade, Serbia; 5Clinic of Infectious and Tropical Diseases, Clinical Centre of Serbia, 11000 Belgrade, Serbia; nevena.todorovic.1992@gmail.com; 6Department of Medical Sciences, Serbian Academy of Sciences and Arts, 11000 Belgrade, Serbia; 7Institute of Medical Statistics and Informatics, 11000 Belgrade, Serbia

**Keywords:** neurological manifestations, long-COVID, antioxidant enzymes, glutathione transferases, polymorphisms

## Abstract

Understanding the sequelae of COVID-19 is of utmost importance. Neuroinflammation and disturbed redox homeostasis are suggested as prevailing underlying mechanisms in neurological sequelae propagation in long-COVID. We aimed to investigate whether variations in antioxidant genetic profile might be associated with neurological sequelae in long-COVID. Neurological examination and antioxidant genetic profile (SOD2, GPXs and GSTs) determination, as well as, genotype analysis of Nrf2 and ACE2, were conducted on 167 COVID-19 patients. Polymorphisms were determined by the appropriate PCR methods. Only polymorphisms in GSTP1AB and GSTO1 were independently associated with long-COVID manifestations. Indeed, individuals carrying *GSTP1 Val* or *GSTO1 Asp* allele exhibited lower odds of long-COVID myalgia development, both independently and in combination. Furthermore, the combined presence of *GSTP1 Ile* and *GSTO1 Ala* alleles exhibited cumulative risk regarding long-COVID myalgia in carriers of the combined *GPX1 LeuLeu/GPX3 CC* genotype. Moreover, individuals carrying combined *GSTM1-null/GPX1LeuLeu* genotype were more prone to developing long-COVID “brain fog”, while this probability further enlarged if the *Nrf2 A* allele was also present. The fact that certain genetic variants of antioxidant enzymes, independently or in combination, affect the probability of long-COVID manifestations, further emphasizes the involvement of genetic susceptibility when SARS-CoV-2 infection is initiated in the host cells, and also months after.

## 1. Introduction

Although the main focus regarding the ongoing coronavirus disease 2019 (COVID-19) pandemic has been the acute clinical manifestations of the disease, at this point, it is clear that understanding the sequelae of the disease is of equal, if not of utmost importance [1,2,3]. Indeed, the novel term “post-COVID-19 syndrome” or “long-COVID” has been introduced for those individuals in whom COVID-19 can cause symptoms that last weeks or months after the infection has resolved. Still, there are no precise criteria for this multi-organ disorder, which includes a wide range of clinical manifestations, including pulmonary, cardiovascular, immunological, neurological, hematologic, endocrine, renal, gastrointestinal, dermatologic and/or psychiatric disorders [4,5,6,7]. Interestingly, according to the National Health System of the United Kingdom, the odds of having long-term symptoms do not seem to be linked to the severity of acute COVID-19 [8].

It has been shown that the prevalence of neurological symptoms in the acute and subacute COVID-19 is up to 90%, varying from sense of smell and taste disorders to headaches, myalgias, cognitive impairment and memory problems, even comprising encephalopathy, delirium, cerebrovascular disease, seizures and neuropathy in some cases [9,10,11,12,13,14]. Manifestations such as abnormal movements, psychomotor agitation, syncope and autonomic dysfunction seem to be less frequent in COVID-19 patients [1,3]. These neurological disorders might be explained by a myriad of ethological factors, including inflammation, cytokine storm, cerebral hypoxia and hypoperfusion [15,16,17,18]. Namely, even mild inflammation may disrupt the blood–brain barrier, possibly even initiating protein aggregation [17,18].

In the course of the acute phase of COVID-19, significant alterations in redox homeostasis also contribute to the vicious cycle of inflammation and oxidative stress [19,20,21,22]. Based on this premise, we have recently recognized the role of the antioxidant genetic profile in the susceptibility and severity of acute COVID-19. Apart from superoxide dismutases (SODs) and glutathione peroxidases (GPXs), a superfamily of glutathione transferases (GSTs) also contributes greatly to antioxidant defense. These multifunctional enzymes are involved in a number of catalytic and non-catalytic processes, however primarily recognized as phase II cellular detoxification system enzymes [23,24]. Importantly, certain GSTs, precisely GSTO1, exhibit a pro-inflammatory role by modulating the pro-inflammatory lipopolysaccharide (LPS)/Toll-like receptor (TLR-4)-induced activation of the nuclear factor kappa B (NF-B) pathway in macrophages [25,26]. Furthermore, these glutathione (GSH)-dependent enzymes demand this important thiol for their function, thus modulating GSH homeostasis. Interestingly, in the case of COVID-19, it has been shown that the severe form of the disease is triggered by conditions leading to decreased glutathione levels [27], which might also be affected by polymorphisms in GSH-dependent enzymes and their impact on glutathione utilization. Indeed, we found that the polymorphisms in *GSTP1*, *GSTM3, GSTO1* and *GSTO2* genes were associated with significant odds of COVID-19 development, while the combined *GSTP1* and *GSTM3* genotype even exhibited cumulative risk regarding both COVID-19 occurrence and COVID-19 severity [26,28]. Furthermore, we have shown the influence of *SOD2* and *GPX1* polymorphisms on the inflammation and coagulation parameters in COVID-19 patients [29].

In long-COVID, numerous data indicate toward an unexpectedly high prevalence of prolonged neurological symptoms among convalescent patients, which are of special interest due to their potential to be life altering [1,2,30]. Fatigue, cognitive impairment and “brain fog”, myalgia, headache, mood disorders, sensorimotor deficits, as well as, smell or taste disorders are among the most frequent neurological sequels of COVID-19 [31]. It is important to note that even in patients in whom neurological symptoms were absent in the acute phase of the disease, microstructure changes, as well as, cerebral blood flow changes were observed at 3-month follow-up [32]. Although mechanisms underlying the manifestation of long-COVID are still unclear, neuroinflammation and disturbed redox homeostasis are suggested as prevailing in neurological sequelae propagation [1]. Indeed, numerous neurological diseases are characterized by disturbances in redox homeostasis, while in some of them (e.g. Parkinson’s disease, Alzheimer’s disease, amyotrophic lateral sclerosis, and epilepsy) polymorphisms occurring in antioxidant genes are even recognized as risk factors [33,34,35,36,37]. In this line, transcription factor Nrf2 (nuclear factor (erythroid-derived 2)-like2), one of the key regulators of cellular redox state that modulates the expression of a large number of genes, including antioxidant enzymes, might also be an important contributing factor [38,39]. Namely, Kelch-like ECH-associated protein 1 or Keap1/Nrf2 pathway reacts to changes in cellular levels of reactive oxygen species (ROS). When ROS levels are increased, Keap1 undergoes allosteric changes that lead to decreased proteasomal degradation and accumulation of Nrf2. Its accumulation, followed by nuclear translocation, results in intensified transcription of Nrf2 target genes [38,39]. Interestingly, the role of Nrf2 has also been investigated in relation to the expression of genes involved in inflammasome assembly, as well as in inflammasome signaling [40,41].

Since there is limited evidence on the pathophysiological mechanisms implicated in the manifestation of long-COVID, and oxidative stress is supposed to be one of them, we aimed to investigate whether variations in genetic profile, especially in genes encoding immediate (superoxide dismutase) and first-line defense (glutathione transferases and glutathione peroxidases) antioxidant enzymes might be associated with neurological sequels in long-COVID.

## 2. Materials and Methods

The study was conducted on 167 COVID-19 patients (100 men and 67 women, with an average age of 55.9 ± 12.3 years) treated in the Clinical Centre of Serbia, between July 2020 and February 2021. All participants were Caucasians by ethnicity. Positive SARS-CoV2 reverse transcription (RT)-PCR test performed from nasopharyngeal and oropharyngeal swabs according to World Health Organization guidelines and using available RT-PCR protocols, age (≥18 years old) and willingness to provide written informed consent were the inclusion criteria for participation in the study.

The principles of the International Conference on Harmonization (ICH) Good Clinical Practice, the “Declaration of Helsinki,” and national and international ethical guidelines were followed during this study with approval obtained from the Ethics Committee of the Clinical Centre of Serbia. Clinical, demographic and epidemiological data were collected using the RedCap^®^ based questionnaire (RedCap^®^. Available online: https://redcap.med.bg.ac.rs/, AntioxIdentification, accessed on 21 February 2022).

The data on medical history, signs and symptoms of the disease, comorbidities and laboratory parameters were obtained from the patients’ clinical records. Follow-up neurological examination conducted 3 months after acute phase of COVID-19 included: evaluation of mental status (orientation and recent memory), examination of cranial nerves by assessment of pupils for size, symmetry and reactivity to light, primary eye position, motor (tone, signs of rigidity and spasticity) and sensitivity response, deep tendon reflexes and pathologic reflexes (Babinski sign), neck stiffness, coordination testing, nystagmus, tremor, assessment of ataxia and gait. The presence of myalgia, fatigue and “brain fog” was also investigated. Precisely, myalgia was evaluated by static and dynamic manual palpation of soft tissue and joints, as well as, the presence of muscle pain during movement. Fatigue was evaluated using the Fatigue Assessment Scale (FAS), which represents a 10-item general fatigue questionnaire. It is based on 10 statements that refer to how individuals usually feel, where individuals answer this 10-item scale using a five-point scale ranging from 1 (“never”) to 2 (“sometimes”), 3 (“regularly”), 4 (“often”) or 5 (“always”). A total FAS score of < 22 indicates no fatigue, while a score of ≥ 22 indicates fatigue [42]. Regarding ”brain fog“, it was evaluated based on individuals’ agreement (agree or disagree) with a list of 19 brain fog descriptors (e.g. forgetful, difficulty thinking and focusing, slow, sleepy, etc.). Agreement in 10 or more descriptors was considered as presence of ”brain fog“ [43]. DNA isolation was performed from EDTA-anticoagulated peripheral blood obtained from the study participants using PureLink™ Genomic DNA Mini Kit (ThermoFisher Scientific, Waltham, MA, USA).

*GSTM1* and *GSTT1* deletion polymorphisms were determined by multiplex PCR, using the *CYP1A1* gene as an internal control. *GSTA1* rs3957357 and *GPX1* rs1050450 polymorphisms were determined using PCR restriction fragment length polymorphism (PCR-RFLP). Primer sequences of *GSTM1, GSTT1, CYP1A1, GSTA1* and *GPX1* genes and PCR protocols details were as previously described [28,29]. *GSTP1* rs1695, *GSTP1* rs1138272, *GSTM3* rs1332018, *GSTO1* rs4925, *GSTO2* rs156697, *ACE2* rs4646116, *ACE2* rs143936283, *SOD2* rs4880 and *GPX3* rs8177412 polymorphisms were determined by real-time PCR onMastercycler ep realplex (Eppendorf, Germany), using TaqMan Drug Metabolism Genotyping assays (Life Technologies, Applied Biosystems, United States). Assays’ IDs were as follows: C_3237198_20, C_1049615_20, C_3184522_30, C_11309430_30, C_3223136_1_, C_32336201_30, C_175425292_10, C_8709053_10 and C_2596717_20, respectively). Nrf2 rs6721961 polymorphism was determined by confronting 2-pair primers (CTPP) PCR method [44]. PCR products were separated on 2% agarose gel stained with SYBR^®^ Safe DNA Gel Stain (Invitrogen, United States) and visualized on Chemidoc (Biorad, United States).

Statistical data analysis was performed using IBM SPSS Statistics 22 (SPSS Inc., Chicago, IL, United States). Results were presented as *n* (%) and mean ± SD. Pearson’s chi-squared test or Fisher’s exact test were used to test for the association between acute COVID-19 manifestations regarding the long COVID-19 sequel. We used univariate logistic regression for calculating the odds ratio (OR) and 95% confidence interval (95%CI) in order to determine the potential association between assessed genotypes and odds for the development of long-COVID. Due to unfavorable relations between the number of outcomes and potential predictors, multivariable analysis was not conducted. All *p*-values less than 0.05 were considered significant.

## 3. Results

The clinical characteristics of 167 convalescent COVID-19 patients included in this study are summarized in Table 1.

As presented, the average BMI indicates that the majority of patients were overweight and 47% were smokers sometime during their lifetime. Regarding the most frequent COVID-19 comorbidities, 35% of patients had hypertension and 11% had diabetes. During the acute phase of COVID-19 febricity (oral temperature exceeded 37.2 °C) was observed in 86% of patients, while 49% had a temperature over 38 °C (fever). Almost all patients who presented with pneumonia were hospitalized (86%) with 38% requiring O_2_ support.

Out of a total number of COVID-19 patients, around 31% reported the loss of smell and taste during the acute phase of the disease. Weakness and general malaise were present in approximately 90% of the patients, while 44% presented with myalgia. Almost half of the COVID-19 patients (46%) suffered from headaches, while in 40%, the headache was associated with fever (Table 2).

Neurological manifestations evaluated during follow-up included fatigue, which was present in 80% of patients, myalgia in 28%, “brain fog” in 13%, as well as instability and paresthesia, present in 9% and 10%, respectively (Table 3). Further analysis showed that 42 COVID-19 patients had both fatigue and myalgia, 20 had both fatigue and “brain fog”, while 6 that had both myalgia and “brain fog” also had fatigue.

The distribution of analyzed genotypes among COVID-19 patients is summarized in Table 4. As presented, genetic variations were assessed for 11 antioxidant enzymes, including SOD2, GPX1 and GPX3, as well as, eight members of the GSTs family. Furthermore, polymorphic expressions of the key redox-sensitive transcription factor Nrf2 and, in addition, two functional polymorphisms in ACE2 were also evaluated (Table 4). As visible, the frequencies of certain genotypes differ from those generally observed in the healthy Caucasian population, especially in the case of *GSTM1, GSTP1AB* and *GSTP1CD, GSTO1, GSTO2* and *GPX3,* as previously described [26,28,29].

Figure 1A–C depict the distribution analysis of acute COVID-19 clinical manifestations assessed in the population of patients with the most prominent long-COVID neurological manifestations, including fatigue, myalgia and “brain fog”, which were further separately analyzed.

Long-COVID fatigue was present in 134 (80%) COVID-19 patients. The majority of them were hospitalized (88.1%), had febricity (88.8%), suffered from COVID-19 pneumonia (92.5%) and reported general malaise (95.5%) or feeling of weakness (96.3%) in the acute phase of the disease (Figure 1A). As presented, pneumonia, feeling of weakness, general malaise and febricity over 38 °C were significantly more frequent (*p* = 0.047, *p* < 0.001, *p* < 0.001 and *p* = 0.043, respectively) in patients with long-COVID fatigue in comparison to those who lacked this manifestation.

Regarding long-COVID myalgia, it was present in 46 (28%) COVID-19 patients, among which the vast majority (84.4%) were hospitalized during the acute phase of COVID-19. These patients presented with COVID-19 pneumonia (91.3%), febricity (93.5%), weakness (97.8%) and general malaise (95.7%) (Figure 1B). When compared to COVID patients without long-COVID myalgia, headache with or without fever and myalgia were significantly more frequent in patients who developed this long-COVID manifestation (*p* = 0.039, *p* = 0.014 and *p* < 0.001, respectively) in the acute phase of the disease.

Interestingly, all 21 (13%) patients experiencing long-COVID-19 “brain fog” suffered from COVID-19 pneumonia and felt weak during the acute phase of the disease (100%). Almost all of them were hospitalized (90.5%), febrile (95.2%) and felt general malaise (95.2%) as well (Figure 1C).

In the next step, a possible association between genetic variability and the probability of developing neurological manifestations in long-COVID were analyzed in the study group, see Figure 2A–C.

Although long-COVID fatigue represents the most frequent neurological manifestation of long-COVID in our study, none of the analyzed fourteen genes was found to significantly affect the odds of its development.

On the other hand, regarding the long-COVID myalgia, we observed that the carriers of at least one *GSTP1AB Val* allele were more than twice less prone to this neurological manifestation (OR = 0.43, 95%CI 0.21–0.87). Similarly, the *GSTO1Asp* allele decreased the odds of long-COVID myalgia nearly two times (OR = 0.51, 95%CI 0.25–1.02) (Figure 2B). What is more, in individuals carrying combined *GSTP1ABValVal/GSTO1 AspAsp* genotype, the odds of having myalgia were even lower (OR = 0.24, 95%CI 0.09–0.66) when compared to carriers of combined *GSTP1IleIle/GSTO1AlaAla* genotype. On the contrary, genetic variability in glutathione peroxidases seems to increase the odds of developing long-COVID myalgia. Namely, *GPX1 Leu* and *GPX3 CC* alleles are associated with increased odds (Figure 2B), which was more than threefold in individuals carrying combined *GPX1LeuLeu/GPX3CC* genotype (OR = 3.06, 95%CI 1.07–8.77). Furthermore, when all potentially “increased-odds related genotypes” for long-COVID myalgia were analyzed together, it was shown that the carriers of combined *GSTP1ABIleIle/GSTO1AlaAla/GPX1LeuLeu/GPX3CC* genotype had 10-fold increased odds (OR = 10.5, 95%CI 1.67–66.09) of having this neurological manifestation in comparison to individuals with *GSTP1ABValVal/GSTO1AspAsp/GPX1ProPro/GPX3TT* genotype (data not shown).

One more neurological sequelae of COVID-19 seems to be affected by variations in genes encoding antioxidant proteins. Namely, *GSTM1-null* genotype and *GPX1 Leu* alleles independently increase the odds of “brain fog” manifestation in long-COVID more than twice (Figure 2C). This effect is even more potentiated when these genotypes are present in combination, reaching almost 13-fold increased odds (OR = 12.98 (95% CI 1.56–1693.46) in individuals carrying *GSTM1-null/GPX1LeuLeu* genotype in contrast to carriers of *GSTM1-active/GPX1ProPro* genotype. Similarly, Nrf2 polymorphism individually increases the odds of long-COVID “brain fog” by approximately 50%. However, when analyzed in combination with GSTM1 and GPX1 genotypes, it seems that the presence of the combined *GSTM1-null/GPX1LeuLeu/Nrf2AA* genotype yields more than 15 times increased odds (OR = 15.55, 95%CI 1.56–2102.22) for this neurological manifestation (data not shown).

## 4. Discussion

Lately, both scientists’ and clinicians’ attention has been raised towards understanding why some people develop long-COVID. Oxidative stress, due to impaired expression of antioxidant enzymes, as well as cytoprotective proteins under the control of the antioxidative response element in the DNA, has been suggested as one of the molecular basis of long-COVID [45,46]. Furthermore, apart from the role in the maintenance of the intracellular redox state, the key redox-sensitive transcription factor Nrf2 has been associated with inflammatory response and inflammasome formation in acute COVID-19, proposing its role in long-COVID development too. Therefore, we speculated that genetic variations in antioxidant enzymes, including different members of the GST family, as well as Nrf2, might modulate individual susceptibility towards the development of long-COVID neurological manifestations.

Among the fourteen polymorphisms analyzed in this study, only GSTP1AB and GSTO1 were found to be independently associated with long-COVID manifestations. Indeed, the data obtained showed that individuals carrying GSTP1 Val or GSTO1 Asp allele exhibit lower odds of long-COVID myalgia development, which was even more potentiated in individuals carrying both of these alleles. On the other hand, the presence of GSTP1 Ile and GSTO1 Ala alleles exhibited cumulative risk regarding long-COVID myalgia in carriers of combined GPX1LeuLeu/GPX3CC genotype. This is in line with the fact that individuals carrying combined GPX1 Leu and GPX3 C alleles have an increased probability of developing this long-COVID manifestation. In addition, one particular genotype combination seems to be associated with another long-COVID neurological manifestation. Precisely, individuals carrying combined GSTM1-null/GPX1LeuLeu genotype are more prone to developing long-COVID “brain fog”, while probability further enlarges if the Nrf2 A allele was also present. To our knowledge, this is one of the first investigations that addressed the association of antioxidant genetic profile and long-COVID manifestations.

Long-COVID, or post-acute sequelae of SARS-CoV-2, refers to “signs and symptoms that develop during or after an infection consistent with COVID-19, which continue for more than 12 weeks and are not explained by an alternative diagnosis” [45]. The proposition that a multitude of organs might be responsible for long-COVID [47] is in line with the fact that COVID-19 is regarded as an endothelial disease, since all organs depend on perfusion by the vascular microcirculation, precisely those capillaries that are composed of endothelium and pericytes [48,49,50]. Since both of these cell types express the ACE-2 protein, after binding to its receptor, SARS-CoV-2 may cause vascular injury on one side, but can also cause blood clot formation in blood vessels both in the periphery and brain [51,52]. Indeed, there is evidence of severe hypoxia in tissues surrounding endothelial cells undergoing apoptosis [53]. This multi-organ disease includes, among others, neurological manifestations that seem to be present in nearly one-third of COVID-19 patients within the first 6 months following acute COVID-19 [31]. In the case of neurological sequelae of COVID-19, apart from capillary dysfunction and hypoxia, impaired glucose metabolism, as well as neuro-inflammation and disturbed redox homeostasis are suggested as a main pathophysiological basis [1,53,54].

Regarding redox disbalance, the sequence of events once SARS-CoV-2 enters the host cells seems to be clear now. Namely, the virus alters biochemical processes and, consequently, causes the overproduction of reactive oxygen species (ROS) and oxidative stress. This further causes mitochondrial dysfunction and DNA damage, but at the same time, it affects the expression of the key redox-sensitive transcription factor Nrf2 [45,55]. Nrf2 deficiency decreases the ability of tissue to properly react to disturbances in redox homeostasis, since many genes targeted by Nrf2 encode enzymes essential in antioxidative stress response [41,56]. Since Nrf2 is also shown to down-regulate the expression of genes involved in inflammasome assembly, including NLRP3, caspase 1, IL-1, and IL-18, in that way inhibiting NLRP3 inflammasome activity [40], this further contributes to the process of viral-induced inflammation and the release of proinflammatory cytokines and other molecules that amplify the innate immune response and lead to cell death [45,46]. However, the question remains whether, and in which way, this sequence of events in acute infection explains clinical manifestations in long-COVID?

It is well established that SARS-CoV-2 binds to the host cell and internalizes using a surface ACE2 protein [57,58]. It has further been speculated that genetic polymorphism in the ACE2 gene and the disruption of ACE2 protein 3D structure could influence the virus-ACE2 interaction, consequentially affecting the viral load and patients’ response to the infection in terms of disease progression and severity [57,59]. Our results suggest that the ACE2 genotype might also be associated with the odds of developing long-COVID manifestations. Namely, the missense ACE2 rs4646116 SNP analyzed in our study, shown to destabilize ACE2 protein [59], seems to decrease the odds of fatigue, myalgia and “brain fog” in long-COVID patients. In this line, certain ACE2 variants, which are regarded as COVID-19-susceptible or -protective in the acute COVID-19, might also be analyzed in terms of susceptibility to long-COVID clinical manifestations.

As suggested, once the virus is in the cell, it disrupts redox homeostasis and triggers activation of redox-sensitive Nrf2 [41,45]. Nrf2 rs6721961 single nucleotide polymorphism, analyzed in our study, is positioned in the middle of the antioxidant response element (ARE) motif and affects the binding of Nrf2 to the ARE. Therefore, individuals carrying the homozygous *AA* genotype exhibit a lower level of Nrf2 mRNA, which further leads to its lower activity [60]. Although Nrf2 polymorphism itself cannot be associated with the probability of long-COVID manifestations in our study, it has been shown that one of its target genes can. Namely, among various enzymes encoded by Nrf2 target genes and regulated by its binding to AREs are glutathione transferases, especially GSTP1 [39,61]. Apart from its role in the detoxification, chemoresistance and antioxidant defense, GSTP1 also possesses binding activity toward macromolecules, as well as small molecules, and acts as a negative regulator of mitogen-activated protein kinase (MAPK)-signaling pathway by forming protein–protein interactions with JNK1 (c-Jun NH2-terminal kinase) [24,62,63]. Interestingly, MAPK is among the signaling pathways activated once SARS-CoV-2 binds ACE2, which potentially elicits the cytokine storm, identified as the major cause of tissue injury. In this line, MAPK inhibitors are recognized as a novel therapeutic approach for modulation of pro-inflammatory cytokine production [64], which might suggest the possible modulatory effect of GSTP1 as well. Additionally, GSTP1 has the potential to form a GSTP1/Nrf2 protein complex, suggesting a possibility that GSTP1 protein might help Nrf2 stabilization and its further actions [65]. So far, GSTP1 variant alleles have been shown to affect the susceptibility and severity of COVID-19. Namely, the GSTP1 Val allele, which is shown to decrease the odds of developing severe COVID-19 [28], decreases the probability of developing long-COVID myalgia in the present study. The frequent occurrence of myalgia, as a clinical manifestation in both acute and long COVID-19, represents a consequence of several interconnected molecular events. Due to the systemic inflammatory response observed in COVID-19 patients, the involvement of upregulated proinflammatory cytokines, especially IL-6 in hyperalgesia, a stated characteristic of myalgia, has been anticipated [66]. Moreover, hypoxic ischemia in the musculoskeletal system associated with hyperlactemia results in diminished ATP production, leading to pain and fatigue. In that context, myalgia could reflect the state of systemic chronic inflammation along with immunological dysfunction observed in long-COVID-19 [67]. In this study, apart from the *GSTP1 Val* allele, the *GSTO1 Asp* allele also exhibits a modifying effect on the probability of the occurrence of myalgia in long-COVID patients, which is even more potentiated in individuals carrying both of these alleles. GSTO1 exhibits diverse sets of functions involved in redox-sensitive signaling pathways and innate immune response [68]. In the first phase of antiviral host defense, a supporting role of GSTO1-1 in the activation of NLRP3 inflammasome has been established [69]. The NLRP3 inflammasome activation is a necessary step for the conversion of pro-IL-1β and pro-IL-18 into respective mature forms, resulting in subsequent IL-6 release [70,71]. Since the *GSTO1 Ala* allele is more efficient in NEK7/NLRP3 inflammasome activation, it is reasonable to speculate that the presence of this GSTO1 allele with higher deglutathionylation activity could represent the underlying mechanism for the associations between this genetic polymorphism and innate immune response [63,72]. Therefore, our results are in accordance with the recent data on highly activated innate immune cells in patients with long-COVID [73].

Long-COVID myalgia can also be associated with genetic variability in glutathione peroxidases. Namely, we have shown that individuals carrying combined *GPX1 Leu* and *GPX3 C* alleles have an increased probability of developing this long-COVID manifestation, while cumulative odds are even higher in a combination of *GSTP1 Ile* and *GSTO1 Ala* alleles. Apart from their significant role in antioxidant defense by preventing detrimental damage caused by H_2_O_2_ [74], GPX1 has been shown to affect ASK1 activation and interact with TRAF2, further interfering with the formation of the active ASK1 complex [75]. This way, similarly to GSTP1, GPX1 also seems to participate in signaling cascades activated by SARS-CoV-2, with possible effects on long-COVID development [76]. Another glutathione peroxidase analyzed in our study, GPX3, has just recently been investigated in critically ill COVID-19 patients as an important factor in the interplay between inflammation and oxidative stress [77]. Due to the fact that infectious diseases are associated with decreased Se levels and reduced GPX activity, while GPX3 seems to be able to mediate the anti-inflammatory effects in case of GPX1 loss [78], the role of this antioxidant enzyme in COVID-19 has emerged, and Se and Zn supplementation trials in COVID-19 have initiated [79,80].

“Brain fog” in long-COVID is also more probable in the carriers of the *GPX1 Leu* allele. Decreased GPX1 activity is possible due to the fact that the change of proline with leucine (Pro200Leu) in this SNP leads to alterations in the secondary and tertiary structure of GPX1 [81]. The individuals in whom another important antioxidant enzyme, such as GSTM1, is lacking, the odds of “brain fog” are even higher, as observed in this study. What is more, this class of GSTs, precisely the *GSTM1-null* genotype, is associated with an increased risk of inflammatory lung diseases [82]. Therefore, it seems reasonable to assume that apart from contributing to oxidative stress and inflammation in the acute COVID-19, the lack of this enzyme affects long-COVID too. Importantly, Nrf2 genetic variability, precisely the *Nrf2 A* allele, when present in combination with *GPX LeuLeu/GSTM1-null* genotype further increases the probability of long-COVID “brain fog”, highlighting its crucial role in redox homeostasis.

It seems that variations in the human genome, apart from age, gender and chronic diseases, represent an important source of variability in determining both COVID-19 severity in the acute phase, but also long-COVID probability. So far, 13 loci in the human genome that affect COVID-19 susceptibility and severity have been identified [83] while the data on the association between genetic variations and long-COVID are insufficient.

This study has several limitations that need to be addressed. Due to a limited number of patients with different neurological manifestations of long-COVID (fatigue, myalgia or “brain fog”), as well as the fact that some of the patients overlap, a multivariable analysis could not be conducted. Unfortunately, due to the same reason, the modifying effect of comorbidities could not be evaluated either. Therefore, our findings should be regarded more as “hypothesis generating” rather than explanatory, demanding a larger study population. Still, although the sample size is rather small, this study may offer some essential information that could be the base for future longitudinal research.

## 5. Conclusions

Taken together, our results on the association between alterations in antioxidant genetic profile and neurological manifestations in long-COVID are in agreement with the current hypothesis regarding the biological basis and pathophysiological mechanisms in this condition. The fact that certain genetic variants of antioxidant enzymes, independently or in combination, affect the probability of long-COVID manifestations, further emphasizes the involvement of genetic susceptibility both when SARS-CoV-2 infection is initiated in the host cells, and also months after. However, further studies are needed to clarify the exact roles of specific enzymes, as well as, redox-sensitive transcription factors. Our results might even contribute to better identification of potential therapeutic strategies for both prevention and treatment of long-COVID sequelae.

## Figures and Tables

**Figure 1 antioxidants-11-00954-f001:**
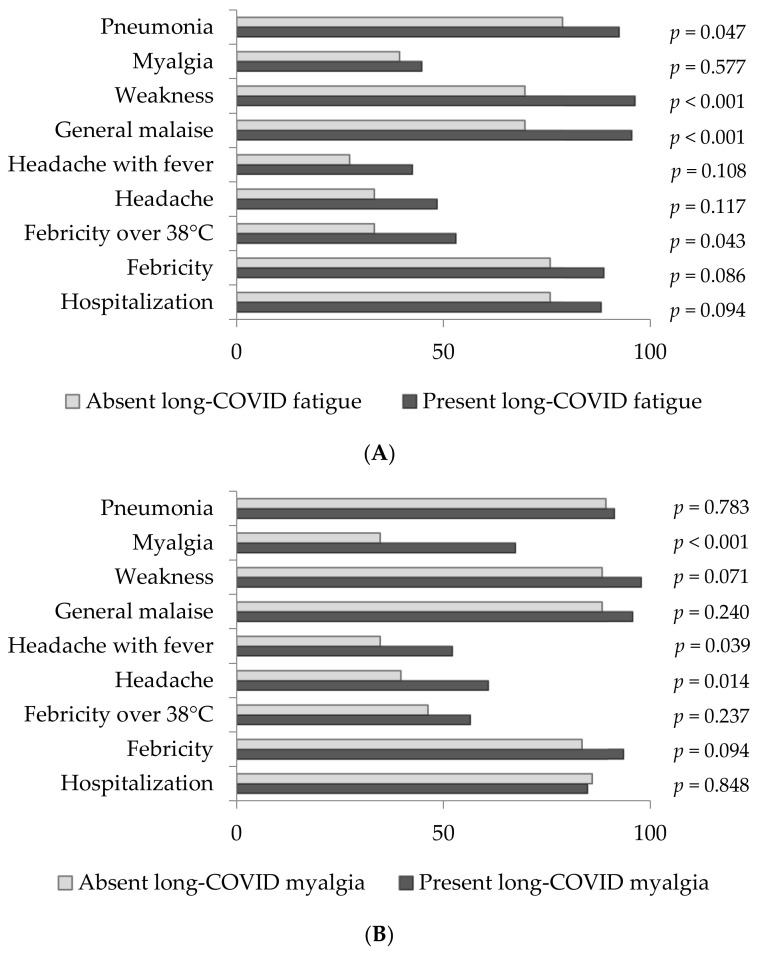
(**A**) The frequency of acute COVID-19 manifestations in population of patients without and with long- COVID fatigue. (**B**) The frequency of acute COVID-19 manifestations in population of patients without and with long-COVID myalgia. (**C**) The frequency of acute COVID-19 manifestations in population of patients without and with long-COVID “brain fog”.

**Figure 2 antioxidants-11-00954-f002:**
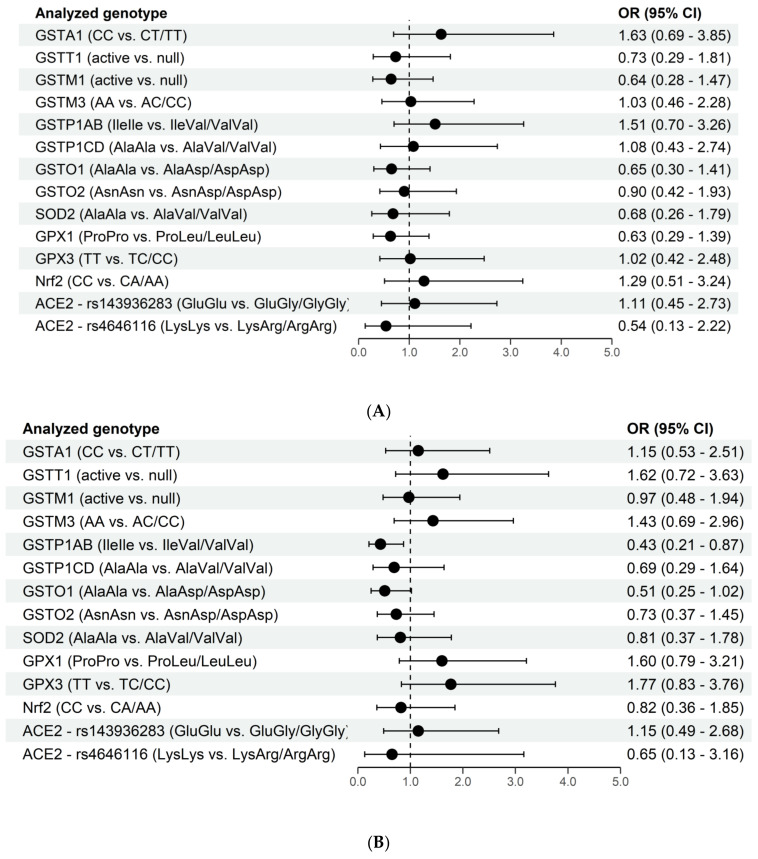
(**A**). The association of genetic variations and long-COVID fatigue. (**B**) The association of genetic variations and long-COVID myalgia. (**C**). The association of genetic variations and long-COVID “brain fog”.

**Table 1 antioxidants-11-00954-t001:** Clinical characteristics of COVID-19 patients.

	COVID-19 Patients
**Age** (years) ^a^	55.9 ± 12.3
**Gender**, *n* (%)	
Male	100 (60)
Female	67 (40)
**BMI** (kg/m^2^) ^a^	29.0 ± 5.0
**Smoking**, *n* (%)	
Never	84 (53)
Former	57 (36)
Ever	17 (11)
**Hypertension**, *n* (%)	
No	109 (65)
Yes	58 (35)
**Diabetes**, *n* (%)	
No	149 (89)
Yes	18 (11)
**Febricity**, *n* (%)	
No	23 (14)
Yes	144 (86)
**Febricity over 38 °C**, *n* (%)	
No	85 (51)
Yes	82 (49)
**Hospitalization**, *n* (%)	
No	24 (14)
Yes	143 (86)
**Pneumonia**, *n* (%)	
No	17 (10)
Yes	150 (90)
**O_2_ support**, *n* (%)	
No	103 (62)
Yes	64 (38)

^a^ Values are presented as mean ± SD; BMI, body mass index.

**Table 2 antioxidants-11-00954-t002:** Neurological manifestations in acute COVID-19.

ACUTE COVID-19	COVID-19 Patients
**Loss of smell**, *n* (%)	
No	115 (69)
Yes	52 (31)
**Loss of taste**, *n* (%)	
No	119 (71)
Yes	48 (29)
**Myalgia**, *n* (%)	
No	94 (56)
Yes	73 (44)
**Weakness**, *n* (%)	
No	16 (10)
Yes	151 (90)
**General malaise**, *n* (%)	
No	15 (9)
Yes	152 (91)
**Headache**, *n* (%)	
No	91 (54)
Yes	76 (46)
**Headache with fever**, *n* (%)	
No	101 (60)
Yes	66 (40)

**Table 3 antioxidants-11-00954-t003:** Neurological manifestations in long-COVID.

LONG-COVID	COVID-19 Patients
**Fatigue**, *n* (%)	
No	33 (20)
Yes	134 (80)
**Myalgia**, *n* (%)	
No	121 (72)
Yes	46 (28)
**“Brain fog”**, *n* (%)	
No	146 (87)
Yes	21 (13)
**Instability**, *n* (%)	
No	152 (91)
Yes	15 (9)
**Paresthesia**, *n* (%)	
No	150 (90)
Yes	17 (10)

**Table 4 antioxidants-11-00954-t004:** Distribution of analyzed genotypes in COVID-19 patients.

Variants	Genotype, *n* (%)	Variants	Genotype, *n* (%)
	** *GSTM1* **		** *GSTT1* **
*active ^a^*	65 (39)	*active ^a^*	131 (79)
*null ^b^*	100 (61)	*null ^b^*	34 (21)
	** *GSTM3 (rs1332018)* **		** *GSTA1 (rs3957357)* **
*AA*	61 (37)	*CC*	52 (37)
*AC*	51 (31)	*CT*	66 (46)
*CC*	53 (32)	*TT*	24 (17)
	** *GSTP1 (rs1695)* **		** *GSTP1 (rs1138272)* **
*IleIle*	82 (49)	*AlaAla*	129 (78)
*IleVal*	66 (40)	*AlaVal*	36 (21)
*ValVal*	18 (11)	*ValVal*	1 (1)
	** *GSTO1 (rs4925)* **		** *GSTO2 (rs156697)* **
*AlaAla*	79 (48)	*AsnAsn*	79 (47)
*AlaAsp*	48 (29)	*AsnAsp*	46 (28)
*AspAsp*	38 (23)	*AspAsp*	41 (25)
	** *Nrf2 (rs672196)* **		** *SOD2 (rs4880)* **
*CC*	124 (75)	*AlaAla*	39 (23)
*CA*	38 (23)	*AlaVal*	89 (54)
*AA*	3 (2)	*ValVal*	39 (23)
	** *GPX1 (rs1050450)* **		** *GPX3 (rs8177412)* **
*ProPro*	75 (45)	*TT*	126 (75)
*ProLeu*	71 (43)	*TC*	39 (24)
*LeuLeu*	20 (12)	*CC*	2 (1)
	** *ACE2 (rs4646116)* **		** *ACE2 (rs143936283)* **
*LysLys*	154 (94)	*GluGlu*	37 (23)
*LysArg*	9 (5)	*GluGly*	126 (77)
*ArgArg*	1 (1)	*GlyGly*	0 (0)

*^a^* Active, if at least one active allele present; ^*b*^ Null if no active alleles present.

## Data Availability

The data are partially contained within the article, while complete data supporting reported results are available at the RedCap platform (Research Electronic Data Capture, Vanderbilt University) of the Faculty of Medicine University in Belgrade.

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
