# Peer review of "Antioxidant Genetic Profile Modifies Probability of Developing Neurological Sequelae in Long-COVID"

_antioxidants, 2022, doi:10.3390/antiox11050954_

Round 1

Reviewer 1 Report

The study from Ercegovac and coll. carries out a gene polymorphism profiling in COVID-19 patients, focusing on antioxidant enzymes, and investigates the correlation with long-COVID symptoms. The study is very interesting and well performed. The introduction and discussion could be improved. I believe this study and further analysis of genetic polymorphisms of antioxidant enzymes would be really useful to build predictive panels for oxidative damage in COVID-19 as well as other inflammatory pathologies, and could be used to prevent long-COVID by a personalized medicine approach employing antioxidant therapeutic strategies. This kind of future perspective could be added in conclusion.

  1. Introduction: a brief description of antioxidant and pro-oxidant function of the analyzed enzymes would be useful. In fact, in several sentences GST are defined defense antioxidant enzymes, however GSTO1 is pro-inflammatory. For example lines 99-100, 258-259, 326.
  2. Introduction: because the study is focused on glutathione-dependent enzymes, briefly illustrating the central role of glutathione in COVID-19 exacerbated inflammation would clarify the reasons to investigate the correlations between these polymorphisms and long-COVID manifestations. For example, referring to the review by Silvagno et al. published in Antioxidants (doi:10.3390/antiox9070624), severe COVID-19 disease is triggered by conditions leading to decreased glutathione levels, these conditions would be the most affected by polymorphisms further impacting on glutathione utilization.
  3. Discussion: because GSTO1 rs4925 polymorphism is correlated to Alzheimer onset, please comment on the lack of correlation with brain fog, and also on possible reasons why GSTO1 combined with different polymorphisms can lead to opposite results on myalgia.
  4. Discussion: the meaning of lines 355-358 is not clear.

Author Response

RESPONSE TO REVIEWER 1: 

  1. “....a brief description of antioxidant and pro-oxidant function of the analyzed enzymes would be useful. In fact, in several sentences GST are defined defense antioxidant enzymes, however GSTO1 is pro-inflammatory.....“

 Indeed, a brief description of antioxidant enzymes, including glutathione transferases, is useful. To comply with the reviewers comment the following text was added to Introduction section, as visible, and References list has been updated accordingly:

Apart from superoxide dismutases (SODs) and glutathione peroxidases (GPXs), a superfamily of glutathione transferases (GSTs) also contributes greatly in antioxidant defense. These multifunctional enzymes are involved in a number of catalytic and non-catalytic processes, however primarely recognized as phase II cellular detoxification system enzymes [23,24]. Importantly, certain GSTs, precisely GSTO1, exhibit pro-inflamatory role by modulating the pro-inflammatory lipopolysaccharide (LPS)/Toll-like receptor (TLR-4)-induced activation of the nuclear factor kappa B (NF-B) pathway in macrophages [25,26].“

  1. “....because the study is focused on glutathione-dependent enzymes, briefly illustrating the central role of glutathione in COVID-19 exacerbated inflammation would clarify the reasons to investigate the correlations between these polymorphisms and long-COVID manifestations. For example, referring to the review by Silvagno et al. published in Antioxidants (doi:10.3390/antiox9070624).....“

We thank the reviewer for this valuable suggestion. The role of glutathione homeostasis is shown to be very important in COVID-19. Therefore, the following thext was added to Introduction section, while suggested refernce was addede to References list:

Furthermore, these glutathione (GSH)-dependent enzymes demand this important thiol for their function, thus modulating GSH homeostasis. Interestingly, in case of COVID-19, it has been shown that severe form of the disease is triggered by conditions leading to decreased glutathione levels [27], which might also be affected by polymorphisms in GSH-dependent enzymes and their impact on glutathione utilization. Indeed,...“

  1. “.... because GSTO1 rs4925 polymorphism is correlated to Alzheimer onset, please comment on the lack of correlation with brain fog, and also on possible reasons why GSTO1 combined with different polymorphisms can lead to opposite results on myalgia.“

It seems that GSTO1, by its impact on oxidative stress and/or inflammation, exhibits multifaceted roles in the pathobiology of different neurological conditions.

 Since GSTO1 is primarily involved in post-translational covalent modifications of proteins via its thioltransferase activity (glutathionylation/deglutathionylation), it seems reasonable that this polymorphism would affect conditions in which structural changes represent one of the underlying mechanisms, such as Alzheimer's disease. Indeed, its significantly lower levels were found in AD hippocampi (Allen, M., Zou, F., Chai, H.S. et al. Glutathione S-transferase omega genes in Alzheimer and Parkinson disease risk, age-at-diagnosis and brain gene expression: an association study with mechanistic implications. Mol Neurodegeneration, 2012). The lack of association between this polymorphism and brain fog might be explained by the fact that brain fog is consequential to different medical conditions, in our case specifically COVID-19, but without a clear underlying mechanism.

GSTO1 is shown to promote activation of the pro-inflammatory cytokine, IL-1β by post-translational processing. In this line, regarding myalgia, the result that GSTO1 varant allele (with lower activation of inflammatory pathways) is associated with lower frequency of myalgia seems logical. However, apart from inflammation, structural changes, but also ischemia are important contributing factors in myalgia. Therefore, polymorphisms in antioxidant genes seem important since endothelium is vulnerable to changes in ROS and RNS concentrations, as well as, their byproducts.

  1. „....the meaning of lines 355-358 is not clear..“

According to the reviewers suggestion, the following text was rephrased and all changes are visible in the text:

Similarly, Nrf2 polymorphism individually increases the odds of long-COVID “brain fog” approximately 50%. However, when analysed in combination with GSTM1 and GPX1 genotypes, it seems that that presence of combined GSTM1-null/GPX1LeuLeu/Nrf2AA genotype yields more than 15 times increased odds (OR=15.55, 95%CI 1.56-2102.22) for this neurological manifestation (data not shown). 

Reviewer 2 Report

The authors explore polymorphisms in antioxidant genes in patients post acute COVID-19 and attempt to correlate identified polymorphisms with symptoms of Long COVID at 3 months following acute illness. Examining the pathobiology of Long COVID is important and the data noteworthy, however a few points need to be addressed below:

Major points

1) In the methods section the authors describe the neurological examination of the patients (Babinskis test etc) in great detail, however the methods for evaluation of 'myalgia', 'fatigue', and 'brain fog' are not described. Was this simply a 'Yes' or 'No' subjective questionnaire or were formal tools used e.g. the Chalder Fatigue Score? How the authors reached their case definitions for 'myalgia', 'fatigue', and 'brain fog' is essential and must be clarified.

2) Regarding Figure 1A-C the authors must clearly define the numbers of patients who were evaluated for each Long COVID criterion against antioxidant polymorphisms. Given the data provided in Table 3, this suggests that for myalgia the numbers evaluated were n=48, for brain fog n=21 and fatigue n=134? The n numbers must be made clear in the text as the bar chart format does not allow the reader to easily discern the numbers of patients for each Figure. The same needs to be clear for Figure 2A-C

3) Regarding both Figure 1A-C and the Figures 2A-C, the number of patients who overlap must be made clear, otherwise there is pseudo or overt duplication i.e. how many patients had Fatigue + Brain Fog or Fatigue + Myalgia or all 3? And were these separately analysed when making the comparisons between those who did not have these symptoms in Figures 1 and 2? If not this must be made clear.

4) My greatest concern lies with the statistical evaluation of the polymorphisms. Usually much larger numbers of patients are required to show any sort of 'significance' in polymorphisms between cases and controls. Indeed evaluation of the largest subgroup ('fatigue' n=134) against the 14 genes in question did not reveal any significance. Therefore it's hard to believe that any significant findings in the smaller groups ('myalgia' n=48 and 'brain fog' n=21) were anything more than just due to chance given the small numbers evaluated. Moreover many of the confidence intervals cross 1 (including some that the authors describe as 'significant'). The authors should highlight that these findings are 'hypothesis generating' rather than providing definitive explanation but that larger numbers would be required to be conclusive. Again the authors need to be very clear about the numbers evaluated as per point 3.

5) A limitations section needs to be included highlighting the above issues: namely such low numbers of patients with myalgia and especially brain fog to make any conclusions about the effect of these polymorphisms, many of the CI cross 1/close to 1, likely that some of the patients overlap e.g. brain fog plus myalgia or fatigue plus brain fog etc.

Minor points:

1) Some features in the clinical table/description seem implausible. It seems strange to me that 86% required hospitalization but 90% had pneumonia. Surely all pneumonia patients would need hospitalization as anyone with consolidation/pneumonia detected on a CXR in the Emergency Room would not be discharged home. Furthermore only 38% needed O2 yet 90% had pneumonia and 86% were hospitalized, surely all hospitalized patients require O2 even if only via nasal cannula? 

2) The term 'febricity' needs to be clarified and how it is distinguished from 'febricity over 38'.

3) The authors note that 35% and 11% of the cohort had hypertension and diabetes, respectively. Is dysfunctional redox balance not also implicated in these comorbidities (especially diabetes) which could impact their ability to respond to COVID-19, independent of their antioxidant polymorphisms? If so, this should be noted. For example in a patient with a 'favorable' antioxidant genetic profile, this may be counteracted by having a comorbidity like diabetes which could still lead to severe acute and/or Long COVID. 

4) In the discussion section the reference supporting the suggestion that sustained endotheliopathy may underpin Long COVID actually describes acute COVID-19. However since then, several papers have been published describing endotheliopathy in Long COVID, including: Fogarty et al, Persistent endotheliopathy in the pathogenesis of long COVID syndrome, J Thromb Haemost, 2021 and von Meijenfeldt et al, Blood Advances 2021. Sustained pro-thrombotic changes in COVID-19 patients 4 months after hospital discharge, Blood Advances 2021. Inclusion of these references is more appropriate in the context of Long COVID.

Author Response

RESPONSE TO REVIEWER 2: 

Major points

  1. „In the methods section the authors describe the neurological examination of the patients (Babinskis test etc) in great detail, however the methods for evaluation of 'myalgia', 'fatigue', and 'brain fog' are not described........case definitions for 'myalgia', 'fatigue', and 'brain fog' is essential and must be clarified.“

 We thank the reviewer for this comment. Indeed, the methods for evaluation of 'myalgia', 'fatigue', and 'brain fog' were not described in detail. To comply with the reviewers comment, the following text was added to the Methods section and the References list was updated accordingly:

Presence of myalgia, fatigue and “brain fog” was also investigated. Preciselly, myalgia was evaluated by static and dynamic manual palpation of soft tissue and joints, as well as, the presence of muscle pain during movement. Fatigue was evaluated using the Fatigue Assessment Scale (FAS) which represents a 10-item general fatigue questionnaire. It is based on a 10 statements which reffer to how individuals usually feel, while individuals answer to this 10-item scale using a five-point scale ranging from 1 (“never”) to 2 (“sometimes”), 3 (“regularly”), 4 (“often”) or 5 (“always”). A total FAS score of < 22 indicates no fatigue, while a score of ≥ 22 indicates fatigue [42]. (Michielsen HJ, De Vries J, Van Heck GL. Psychometric qualities of a brief self-rated fatigue measure: The Fatigue Assessment Scale. J Psychosom Res, 2003; 54(4):345-52. doi: 10.1016/s0022-3999(02)00392-6.). Regarding „brain fog“, it was evaluated based on individuals agreement (agree or disagree) with a list of 19 brain fog descriptors (eg. forgetful, difficulty thinking and focusing, slow, sleepy...). Agreement in 10 or more descriptors was considered as presence of „brain fog“ [43]. (Ross AJ, Medow MS, Rowe PC, Stewart JM. What is brain fog? An evaluation of the symptom in postural tachycardia syndrome. Clin Auton Res, 2013; 23(6): 305–311.)“

All changes were made using the track changes option, hence are visible in the text.

  1. Regarding Figure 1A-C the authors must clearly define the numbers of patients who were evaluated for each Long COVID criterion against antioxidant polymorphisms......“

 To comply with the reviewers comment, the appropriate changes were made in the Results section, so that the numbers of patients with myalgia, fatigue and brain fog are clear. All changes are visible in the text due to the track changes option which was used:

Long-COVID fatigue was present in 134 (80%) COVID-19 patients.“

„Regarding long-COVID myalgia, it was present in 46 (28%) COVID-19 patients, among which the vast majority....“

„Interestingly, all 21 (13%) patients experiencing long-COVID-19 “brain fog......”

  1. „Regarding both Figure 1A-C and the Figures 2A-C, the number of patients who overlap must be made clear, otherwise there is pseudo or overt duplication“

 Indeed, clarifying how many patients had 2 or 3 long-COVID manifestations simultaneously is useful and important. Therefore, the following text was added to the Manuscript, section Results:

Further analysis showed that 42 COVID-19 patients had both fatigue and myalgia, 20 had both fatigue and “brain fog”, while 6 that had both myalgia and “brain fog” also had fatigue.”, as visible in the text due to the track changes option.

Furthermore, to clearly state that these were separately analysed, the following text was also added to Results section, as visible:

„Figure 1 (A, B and C) depicts the distribution analysis of acute COVID-19 clinical manifestations assessed in the population of patients with most prominent long-COVID neurological manifestations, including fatigue, myalgia and “brain fog”, which were further separately analyzed.

  1. „...evaluation of the largest subgroup ('fatigue' n=134) against the 14 genes in question did not reveal any significance. Therefore it's hard to believe that any significant findings in the smaller groups ('myalgia' n=48 and 'brain fog' n=21) were anything more than just due to chance given the small numbers evaluated. Moreover many of the confidence intervals cross 1 (including some that the authors describe as 'significant'). The authors should highlight that these findings are 'hypothesis generating' rather than providing definitive explanation but that larger numbers would be required to be conclusive. Again the authors need to be very clear about the numbers evaluated as per point 3.“

We thank the reviewer for this comment. Regarding analysis, it included the whole study group comprising 167 patients, however the number of outcomes varied. Since the relation between the outcomes and potential predictors was unfavourable, only univariate logistic regression could be applied. To clarify this, the following sentence was added to Materials and Methods section.

Due to unfavorable relation between the number of outcomes and potential predictors, multivariable analysis was not conducted.

This fact was also mentioned in limitations section, as visible in the text.

  1. „A limitations section needs to be included highlighting the above issues.....“

Indeed, adding a limitations section is necessary. Therefore the following text was added at the end of Discussion section:

This study has several limitations that need to be addressed. Due to a limited number of patients with different neurological manifestations of long-COVID (fatigue, myalgia or “brain fog”), as well as, the fact that some of the patients overlap, multi-variable analysis could not be conducted. Unfortunately, due to the same reason, the modifying effect of comorbidities could not be evaluated either. Therefore, our findings should be regarded more as “hypothesis generating” rather than explanatory, demanding larger study population. Still, although the sample size is limited, this study may offer some essential information that could be the base for future longitudinal research.

Minor points

 „Some features in the clinical table/description seem implausible. It seems strange to me that 86% required hospitalization but 90% had pneumonia......“

 Indeed, all patients with pneumonia require hospitalization, unless they are without respiratory failure and comorbidities which could lead to severe form of the disease, in which case they are treated via outpatient clinic.

Regarding the percentage of patients requiring O2, the healthcare system in Serbia was organized in a way that all hospitals were in COVID system and even new hospitals were built. Therefore, the vast majority of COVID-19 patients were hospitalized, including those with the mild form of the disease who did not need O2 support. Indeed, according to official data, approximately 20% of hospitalized patients developed a severe form of the disease.

  1. „The term 'febricity' needs to be clarified and how it is distinguished from 'febricity over 38'“

 We completely agree that it needs to be clarified whether the patient had febricity below or over 38°C. Therefore, the following sentence in the Results section has been rephrased:

During acute phase of COVID-19 febricity (oral temperature exceeded 37.2°C)  was observed in 86% of patients, while 49% had temperature over 38°C (fever).

  1. „The authors note that 35% and 11% of the cohort had hypertension and diabetes, respectively. Is dysfunctional redox balance not also implicated in these comorbidities (especially diabetes) which could impact their ability to respond to COVID-19, independent of their antioxidant polymorphisms? If so, this should be noted......“

 We completely agree that chronic conditions, such as hypertension and diabetes, which are  associated with redox disbalans have the impact on COVID-19 and individuals ability to respond to SARS-CoV-2 infection and, later on, develop long-COVID. However, since multivariable analysis could not be conducted, only the potential effect of polymorphisms was evaluated.

To comply with the comment, this is mentioned in the limitations section.  

  1. „....several papers have been published describing endotheliopathy in Long COVID, including: Fogarty et al, Persistent endotheliopathy in the pathogenesis of long COVID syndrome, J Thromb Haemost, 2021 and von Meijenfeldt et al, Blood Advances 2021. Sustained pro-thrombotic changes in COVID-19 patients 4 months after hospital discharge, Blood Advances 2021. Inclusion of these references is more appropriate in the context of Long COVID.“

 To comply with the reviewers comment, suggested refernces were added and the References list was updated accordingly.

Round 2

Reviewer 2 Report

All comments have been addressed satisfactorily with inclusion of limitations and caveats as suggested. Overall I believe the manuscript has improved greatly.